# Assessing the Lifetime Cost-Effectiveness of Low-Protein Infant Formula as Early Obesity Prevention Strategy: The CHOP Randomized Trial

**DOI:** 10.3390/nu11071653

**Published:** 2019-07-19

**Authors:** Diana Sonntag, Freia De Bock, Martina Totzauer, Berthold Koletzko

**Affiliations:** 1Mannheim Institute of Public Health, Social and Preventive Medicine, Mannheim Medical Faculty of the Heidelberg University, 68167 Mannheim, Germany; 2Department of Health Sciences, University of York, Heslington, York YO10 5DDY, UK; 3Federal Centre for Health Education, D-50825 Cologne, Germany; 4Department of Pediatrics, Dr. von Hauner Children’s Hospital, University of Munich Medical Centre, 80337 Munich, Germany

**Keywords:** obesity prevention, cost-effectiveness, markov model, early nutrition, childhood, formula

## Abstract

Background: Although there is a growing number of early childhood obesity prevention programs, only a few of them are effective in the long run. Even fewer reports exist on lifetime cost-effectiveness of early prevention strategies. This paper aimed to assess the lifetime cost-effectiveness of infant feeding modification aiming at reducing risk of later obesity. Methods: The simulation model consists of two parts: (a) Model I used data from the European Childhood Obesity Project (CHOP) trial (up to 6 years) and the German Interview and Examination Survey for Children (KiGGS) (6–17 years) to evaluate BMI trajectories of infants receiving either lower protein (LP) or higher protein (HP) content formula; and (b) Model II estimated lifetime cost-effectiveness based on Model I BMI trajectories. Compared to HP formula, LP formula feeding would incur lower costs that are attributable to childhood obesity across all decades of life. Results: Our analysis showed that LP formula would be cost-effective in terms of a positive net monetary benefit (discounted 3%) as an obesity prevention strategy. For the 19% of infants fed with formula in Germany, the LP strategy would result in cost savings of € 2.5 billion. Conclusions: Our study is one of the first efforts to provide much-needed cost-effectiveness evidence of infant feeding modification, thereby potentially motivating interventionists to reassess their resource allocation.

## 1. Introduction

Childhood obesity has become a major pediatric health concern as prevalence rates have increased substantially in the last few decades. Rates of childhood obesity in Europe, while lower than those in the US and Australia, have increased considerable in the last few decades [1]. This increasing prevalence of overweight and obesity in children is alarming for a number of reasons: immediate health effects, psychological and social consequences of weight stigma, increased costs to already strained primary health care sectors, the persistence of overweight and obesity into adulthood [2], and substantial long-term economic consequences [3,4]. Indeed, the only European pediatric study to date that estimated the long-term economic consequences of childhood obesity found that they totaled €8471 (€9473) for males (females) with obesity in Germany in 2010 [4,5].

With the health and economic consequences of childhood overweight and obesity challenging the sustainability of primary health care systems, there is a growing interest in infant nutrition as an early obesity prevention strategy [6]. While there is increasing evidence of the health benefits of breastfeeding, only a few studies have analyzed the long-term health consequences of improving infant formula nutrition [7,8]. In fact, the European Childhood Obesity Project (CHOP) is one of the first randomized controlled trials that examines the effect of protein intake in formula-fed infants during the first year of life on long-term fat body mass (measured by body mass index (BMI)) [9]. Infants born between October 2002 and July 2004 from uncomplicated singleton pregnancies were enrolled and double-blind randomly assigned to either lower protein (LP) or higher protein (HP) intake groups during the first eight weeks of life in five European countries, including Germany. The formulas differed in the content of cow milk protein (2.05 compared with 1.25 g/dl in infant formula) but had identical energy contents achieved by the adjustment of total fat content. The trial demonstrated that children receiving HP content formula instead of LP formula had a significantly higher BMI and a greater risk of becoming obese at six years of age [9]. Since the CHOP trial reported long-term effectiveness until children enter school, it raises important questions about the potential to save costs over a lifetime. However, no studies in the field of infant feeding include an assessment of costs and effects over lifetime. Yet precisely such an assessment could provide the information needed to reassess and optimize scarce resources in early obesity prevention strategies [10,11]. 

The present study bridges this gap in the literature by extending an established infant feeding modification program (LP versus HP) in two new directions. Firstly, by using a simulation approach, we explored lifetime obesity prevention outcomes based on the data from the original randomized–controlled trial that followed a large cohort of European children from birth to 6 years of age. Secondly, we conducted the first European cost-effectiveness analysis to determine the long-term economic outcomes of the LP strategy (when compared to HP) as an early obesity prevention strategy in Germany.

## 2. Materials and Methods 

### 2.1. Subjects and Methods

In conducting a cost-effectiveness analysis, we extended a modeling approach developed by Heidelberg University in 2017 that estimated the lifetime costs by taking the history of childhood overweight and obesity into account. While this approach [3,4] was limited to lifetime costs of overweight and obesity, our extended simulation model considers lifetime costs and intervention effects on BMI in children using the Markov modeling approach. Markov modeling is a cohort simulation approach commonly used to evaluate long-term risks of morbidity/mortality and associated costs/effects in a cohort [12]. The simulation model and all graphs were developed in Microsoft Excel 2010. While Monte Carlos simulations were programmed in Visual Basic for Application, further statistical analysis was conducted using Stata version 14.1 (StataCorp L.P., College Station, Texas, USA).

### 2.2. Model Structure

Our simulation model consists of two parts: the childhood-to-adolescence model (M1) and the adulthood model (M2). As shown in Figure 1, we created two base cohorts: children fed with LP formula and children fed with HP formula. For both infant feeding schemes, we assumed that children enter M1 at age 0 and move between BMI states annually until the age of 18. Similar to Sonntag et al. [3,4], M1 has the health states (1) normal weight (includes underweight), (2) overweight, (3) obese, and (4) the absorbing state dead. These states are defined according to the age- and sex-specific BMI percentiles of Kromeyer-Hauschild, which are commonly used in Germany: normal weight (≤P90), overweight (>P90 to P97) and obese (>P97) [13]. The initial distribution of the starting cohorts in M1 (independent of whether children are being fed LP or HP) was based on CHOP data (Appendix A) [9]. Moving between BMI states was estimated by annual age- and sex-specific state transition probabilities, which depend on the child being fed LP formula or HP formula (Figure 1). We assumed, consistent with the original CHOP trial, that the intervention effect (represented by a reduced relative risk of becoming overweight/obese if children are fed with LP formula compared to HP formula) was maintained six years.

For both infant feeding schemes, surviving adolescents moved from the childhood-to-adolescence model (M1) to the adulthood model (M2), entering at the BMI stage corresponding to their BMI history. Individuals who were always normal weight in M1 entered M2 in the cohort “always normal weight”, while individuals who were overweight or obese at least once during M1 entered the cohort “overweight/obese at least once during childhood”. Sonntag et al. [3,4] described this model in detail. The BMI states in M2 are defined according to WHO cut-off points (1) normal weight (BMI < 25 kg/m^2^), (2) overweight (25 kg/m^2^ ≤ BMI < 30 kg/m^2^), and (3) obese (BMI ≥ 30 kg/m^2^) [14]. The M2 model also incorporates increases in the risk of morbidity and mortality during adulthood if individuals were overweight/obese during M1. 

Cohorts entering M2 were simulated over lifetime based on transition probabilities for each infant feeding strategy (LP and HP). That is, cohort sizes in each BMI state per cycle were multiplied by age- and sex-specific BMI state costs. Total costs, i.e., costs attributable to obesity/overweight that occurred either in childhood or during adulthood, were calculated by summing up costs over lifetime for each cohort. The sum of the lifetime costs of the cohort “always normal weight” and the cohort “overweight/obese at least once during childhood” provides the estimate of lifetime costs attributable to overweight/obesity. Similarly, lifetime quality-adjusted life years (QALYs) were estimated by multiplying the cohort size in each BMI state per cycle by age- and sex-specific utilities in M2. The incremental costs and QALYs attributable to the different infant feeding groups were then calculated by comparing cost and QALYs in the LP scenario with those in the HP scenario (Figure 1). Additionally, we ran 4000 simulations to estimate the magnitude of costs and QALYs in both scenarios. Commonly used statistical distributions were implemented for relative risks (log normal distribution) and costs (gamma distribution). A uniform distribution was used only when the parameter range was available without standard errors. The 95% confidence intervals based on these simulations are reported in the Results section.

Finally, in overcoming the concerns associated with the incremental cost-effectiveness ratio (ICERs) (e.g., interpretation of negative ICERs [12]), we calculated the net monetary benefit (NMB) by multiplying the gain in health (incremental QALYs) by the willingness to pay (WTP) for the benefit of LP formula and subtract incremental costs. 

### 2.3. Model Parameters

The simulation model was based on data from the CHOP trial (collected in 2010) and the previously established German obesity model [3,4], as described below (Appendix A for all parameters and data sources).

#### 2.3.1. State Transition Probabilities between BMI States

Annual age- and sex-specific transition probabilities for the first age groups (ages birth to 6 years) were based on longitudinal data from the CHOP trial participants [9] and derived by following Miller and Homan [15]. Annual age- and sex-specific transitions for the remaining age groups (ages 7 to 100 years) were taken from Sonntag et al. [3,4], who used data from two sources (Interview and Examination Survey for Children (KiGGs) survey and the German Microcensus 2009). We used the KiGGs survey, a nationally representative prevalence study of 14,747 German children between 0 and 17 years [16], to obtain transition probabilities during childhood and adolescence. The Microcensus data for the year 2009 from the German Federal Statistical Office [17] included detailed age-specific anthropometric measurements for German adults aged 18–100; we used these data to acquire transition probabilities during adulthood. 

#### 2.3.2. Risk of Mortality

Mortality risks during childhood and adolescence were based on age- and sex-specific mortality rates reported in the most recent life tables [18]. Increased relative risk of mortality due to overweight and obesity during adulthood was calculated using the European Prospective Investigation into Cancer and Nutrition [19], as in Sonntag et al. [3,4]. Similarly, to allow for a higher mortality risk during adulthood due to childhood overweight and obesity, we used data from Sonntag et al. [3,4] (Appendix A).

#### 2.3.3. Intervention Costs and Costs Associated with Overweight and Obesity 

Since the cost of producing infant formula with lowered protein content is to our knowledge not higher than that of producing conventional HP formula, our study does not include any costs related to infant feeding [9]. However, we considered direct and indirect costs related to overweight and/or obesity (Appendix A). While direct costs are due to neoplasms, endocrine diseases, cardiovascular, and digestive diseases, indirect costs include lost productivity from paid and unpaid work resulting from sickness absences, early retirement, and early death due to causes attributable to overweight and/or obesity. Both direct and indirect costs were based on a systematic literature review, as in Sonntag et al. [4] (Appendix A). All costs were indexed to the year 2015 in euros (€).

#### 2.3.4. Quality of Life 

We used European Quality of Life-5 Dimensions (EQ-5D) utility indices from Sonntag et al. [20] to calculate QALYs which account for long-standing illnesses, such as diabetes type 2. These were age- and sex-specific and calculated based on a representative sample of the German population [21] using an algorithm by Dolan et al. [22]. To account for a potentially lower quality of life due to overweight and obesity, we took EQ-5D utility indices for overweight and obesity from Sonntag et al. 2016. 

#### 2.3.5. Sensitivity and Scenario Analyses

Sensitivity analyses were applied to test the robustness of our results (Appendix A), while in scenario analyses, we estimated the range of potential cost savings by taking into account that rates of formula feeding are higher in other epidemiological settings, such as the US (30%) [23]. 

## 3. Results

We found that the proportion of individuals with obesity during childhood would be higher if infants were fed HP-content formula, which is in line with previous findings of CHOP at school age (Appendix A) [9]. However, the proportion of individuals with overweight would be substantially higher in the HP group than in the LP one only until school age. Beyond 11 years of age, only marginal differences (1%–2%) would be observed, which would even decrease with age.

Figure 2 evaluates for both feeding groups the lifetime cost per decade, taking the history of childhood obesity into account. It shows that excess weight in childhood seems to be the key contributor to the overall burden of overweight and obesity over a lifetime. Although we did not find a statistically significant difference between the LP formula, compared to the HP formula, the figure indicates that the LP formula could be associated with lower lifetime costs attributable to childhood overweight and obesity. Indeed, compared to children receiving HP formula, children fed with LP formula would have €750 per person (discounted at 3%) fewer lifetime costs attributable to childhood overweight and obesity (Figure 3). 

Figure 3 also shows that children fed with LP formula would spend on average 10 fewer years in overweight and/or obesity states than children receiving HP formula. For instance, feeding a child with LP formula would avert 22 years with overweight or obesity (see red triangle in Figure 3) compared to 12 years if a child was fed with HP formula (see yellow rectangle in Figure 3). This indicates that the LP formula would offer good value for money, as it is associated with higher additional benefits (= fewer years in overweight and/or obesity states) and fewer extra costs, which mainly results from a lower risk of obesity earlier in life. Indeed, this is supported by the results of the cost-effectiveness analysis, which compares lifetime costs attributable to overweight/obesity and outcomes (measured in QALYs) of the LP formula group with those of the HP formula group (= comparator) (Figure 4). Over a lifetime, we predicted for the LP formula 47.72 (95% CI: 47.29;48.02) QALYs and €12535 (95% CI: 6598;13752) costs per person attributable to overweight and/or obesity (discounted at 3%). For the HP formula, we estimated 47.42 (95% CI: 46.94;47.82) QALYs and €13285 (95% CI: 7063;14694) costs per person over a lifetime attributable to overweight and obesity. Therefore, under current modeling assumptions, the LP formula would be cost-effective as an early obesity prevention strategy (expressed as positive NMBs, discounted at 3%). This finding also holds if we run the simulation for males and females, respectively. Given a maximum willingness to pay of €5000 on the part of society for the benefit of the LP formula, there is a 76% chance that the LP formula would be cost-effective; this probability increases up to 85% for a WTP of at least €20000 (Appendix A; for more detailed sensitivity analyses, see Appendix A).

Given that the LP formula would result in slightly higher QALYs and lower lifetime costs attributable to overweight and obesity, it is important to calculate the scale of potential cost savings on the population level if conventional infant formula were replaced by LP formula. We found that infant feeding programs could spend up to €750 per child (€13,285–€12,535; Figure 4) and still yield in positive economic returns. Given that 19% of German infants are formula-fed [24], the LP formula would result, under current modeling assumptions, in lifetime cost savings of €2.5 billion. Moreover, potential cost savings would increase to €4.1 billion if 30% [23] of infants were formula-fed, which is currently the case in the US.

## 4. Discussion

Our study is one of the first efforts to demonstrate that infant formula feeding modification as early obesity prevention would be cost-effective under current modeling assumptions, and this mainly results from a lower prevalence of obesity in early life and a higher life expectancy in the LP group. Moreover, differences in the prevalence of overweight between the LP and the HP group are only observable until school age, which is mainly due to the decay of the small intervention effect of the CHOP trial [9]. These findings are potentially essential information for implementation specialists, which would enable them to better allocate their often-scarce resources to other obesity prevention policies already implemented to modify food environment [25,26]. However, it has to be kept in mind that only a minority of children (19% based on a most recent review) [24] are fed with formula. Breastfeeding is the standard for infant feeding practice and leads to similar effects on BMI and reduced risk of later obesity [9] as does the LP formula. Moreover, despite its lack of flexibility, breastfeeding may incur significantly lower costs, which could make it the most cost-effective strategy to reduce the obesity burden. Countries with low breastfeeding rates, such as the US, France, and Canada [27], could possibly benefit from legislation that prohibits the sale of HP formula and fosters breastfeeding. Indeed, more rigorous EU guidelines, passed in 2006, already restricted the maximum content of cow milk protein in infant formula [28]. 

While most economic evaluations assess the cost-effectiveness of school programs [10,29], there are evidence gaps as to what may be cost-effective as only a few international studies [30,31,32] have analyzed whether early obesity prevention programs offer good value for money. For instance, Wright et al. [32] demonstrated that a US multicomponent child care-based obesity policy intervention, including beverage, physical activity, and screen time regulations, would be cost-saving over a period of ten years. Because of dissimilar health care systems and methodological differences, their results are not directly comparable to ours. Wright et al. [32], for example, estimated long-term costs and effects based on the childhood obesity cost-effectiveness study (CHOICES) framework that models disease-specific pathways. The CHOICE framework differs from our approach of modeling BMI-specific pathways.

Our paper has a number of strengths. To our knowledge, it is the first simulation-based study for Germany that quantifies the long-term cost-effectiveness of an infant feeding modification; thus, it could give methodological guidance for health economic evaluations of early obesity prevention strategies. Another strength of the study is that it is based on data from a large randomized controlled trial with a long original data follow-up period. Indeed, original anthropometric measurements until school entrance allowed a comprehensive and accurate assessment of the intervention effect, which is more valid than assumptions about the maintenance of the intervention effect (as often is done in simulation studies). Moreover, since our main outcome (BMI) was directly measured, a simulated translation of the effects of infant feeding modification into changes in BMI was not necessary, thereby increasing the accuracy of results. Furthermore, our study uses a dynamic simulation approach to estimate the impact of infant feeding modification. This allows an estimation of both short-term consequences (infant feeding modification during the trial) and long-term consequences (BMI and cost trajectories). The substantial long-term consequences of childhood feeding practices that we estimated may underscore the need to prioritize resources for early obesity prevention. 

As the limitations of the modeling approach and the CHOP trial (e.g., measuring BMI rather than adiposity) are extensively discussed in earlier works [3,4,9,33], our focus here is on the limitations of the cost-effectiveness analysis. First, we acknowledge that our study does not capture both the implications of more rigorous EU guidelines [28] passed during the trial period and all potential costs of the intervention and future costs related to obesity, such as losses in human capital and productivity due to reduced fertility rates. Our findings are thus likely to be conservative. In detail, our model does not capture the costs of formula feeding (e.g., cost of formula, cost of equipment, potential additional cost of producing LP). However, since these costs are incurred in both infant feeding strategies, excluding them does not influence the robustness of our results. Moreover, we assessed this concern in the sensitivity analyses, where we evaluated the impact of cost of formula feeding in a low- and high-cost scenario [27]. Second, in the absence of validated preference-based quality-of-life instruments for young children, quality of life was only estimated for adolescents and adults. However, if valid data about quality of life among young children are available, our model can be extended to capture both QALYs and costs during childhood. 

We conclude that under current modeling assumptions, reducing protein content in infant formula would be cost-effective as a potent and sustainable obesity prevention strategy. Indeed, our study is one of the first efforts to demonstrate that such a reduction is not merely an effective obesity prevention strategy from a pediatric and epidemiological perspective but also offers good value for money. With this economic benefit in mind, intervention specialists could be motivated to reassess and optimize the allocation of their resources for early obesity prevention strategies.

## Figures and Tables

**Figure 1 nutrients-11-01653-f001:**
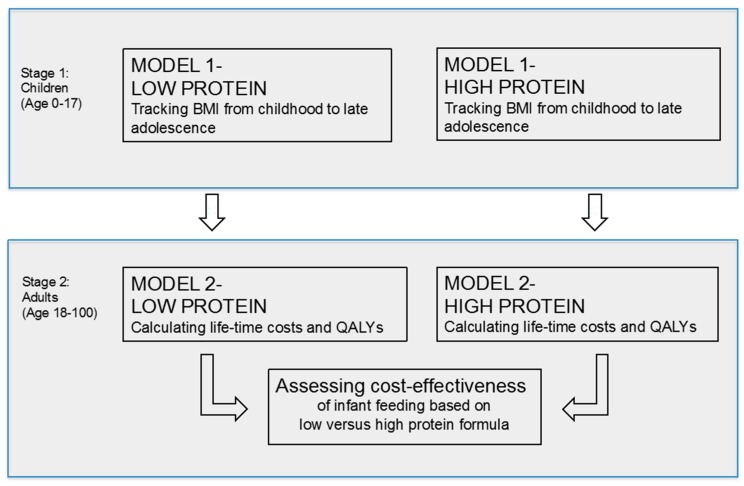
Conceptual model assessing the cost-effectiveness of LP content formula compared to HP one. The simulation model consists of two parts: the childhood-to-adolescence model (M1) and the adulthood model (M2). Two base cohorts were created: children fed with LP formula and children fed with HP formula.

**Figure 2 nutrients-11-01653-f002:**
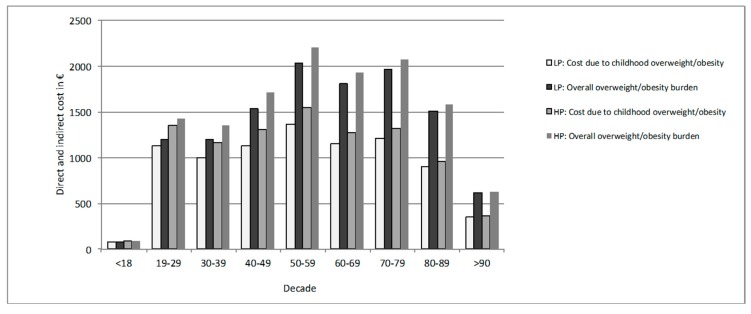
Total lifetime cost attributable to overweight and obesity occurring in different life decades, stratified by infant feeding strategy. Total lifetime cost attributable to overweight and obesity occurring in different life decades as well as partial costs directly attributable to childhood overweight and obesity, stratified depending on whether an infant was fed with LP or HP content formula.

**Figure 3 nutrients-11-01653-f003:**
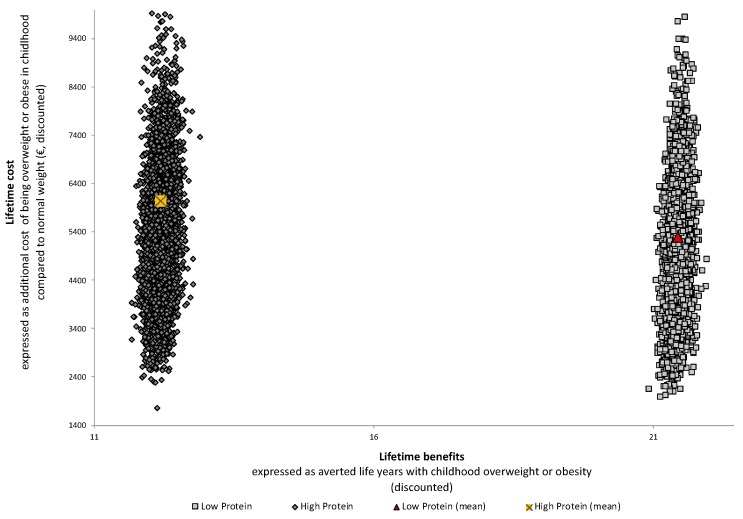
Lifetime excess cost and benefits through childhood overweight and obesity, stratified by infant feeding strategy (lower protein (LP) or higher protein (HP)) based on Monte Carlo simulation, 4000 runs. Lifetime excess costs represents the additional costs of being overweight/obese in childhood compared to normal weight. Lifetime excess benefits are expressed as averted life years with overweight/ obesity.

**Figure 4 nutrients-11-01653-f004:**
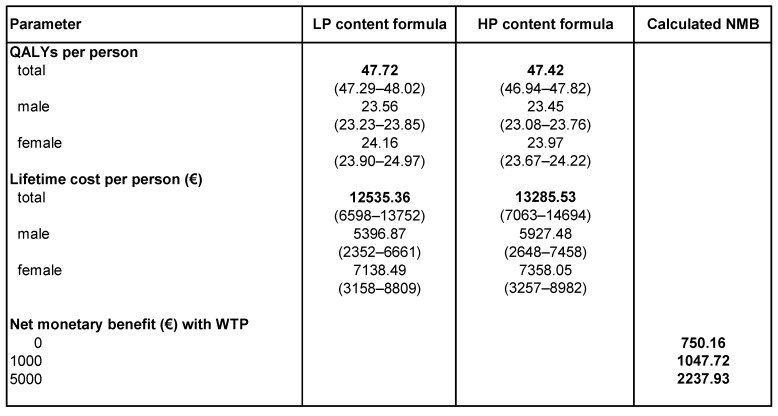
Cost-effectiveness results (2015). Projected cost-effectiveness of LP versus HP content formula, 2015, lifetime QALYs per person, WTP = willingness to pay, NMB = net monetary benefit lifetime total (direct and indirect) costs per person, in parentheses 2.5 and 97.5 sensitivity bounds (Monte Carlo simulation), 4000 simulations, point estimate of NMB, calculated as (incremental QALYs*WTP) incremental costs, NMB for males: €750 (WTP = 0), €1048 (WTP = 1000), €2238 (WTP = 5000), NMB for females: €220 (WTP = 0), €401 (WTP = 1000), €1126 (WTP = 5000); estimated incremental cost–effectiveness ratio (ICER) (€/QALY): lifetime = –2521.10, late adolescence (<20 years) = –1705.92.

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
