# Peer review of "Assessing the Lifetime Cost-Effectiveness of Low-Protein Infant Formula as Early Obesity Prevention Strategy: The CHOP Randomized Trial"

_nutrients, 2019, doi:10.3390/nu11071653_

Round 1

Reviewer 1 Report

This manuscript demonstrated important obesity impact on economy and how a diet intervention early in life could change long term outcome. However, there are a few comments for the authors to address as below:

1)      How about fertility? Obesity affects fertility in both genders, how did the authors take this into account in terms of economic impact?

2)      Can the authors be specific on what the differences is in ‘Beyond 11 years of age, only marginal differences would be observed, which would even decrease with age’?

3)      Should have a line in the methods stating what program was used to draw the figures. It is not advise to use Excel for figures.

4)      In Figure 2, is there any significant difference at all at any time point?

5)      Line 178: which figure is the authors referring to? Figure 2 or 3?

6)      There is no p-value indicated to show significance? There should be a Statistics section in Methods to inform how the data were analysed for differences between groups.

7)      Can the authors discuss in the discussion on why ‘Beyond 11 years of age, only marginal differences would be observed, which would even decrease with age’?

8)      There are a few studies that have showed that rather than BMI per se, it is adiposity that is strongly associate with increased risk of disease development. Is there any adiposity data in the CHOP trial? Otherwise, it would be something for the authors to discuss as limitation.

9)      Is there a reason the authors only compared LP between HP, and not include breastfed in the model?

10)   Was the results different between sexes?

11)   Can the authors elaborate on what type of disease was taken into account in the QALY and life-time costs?

12)   Can the authors generate the male figure for Figure S2?

Author Response

Dear Ms. Duan,

Thank you for your email from June 11 2019 regarding our submission entitled “Assessing the lifetime cost-effectiveness of low-protein infant formula as early obesity prevention strategy: the CHOP randomized trial”. We greatly appreciate the opportunity to revise and resubmit our manuscript for potential publication in Nutrients.

We would like to thank the reviewers for their many helpful comments and suggestions, which have been addressed in the revised manuscript. We feel this has strengthened the manuscript significantly. Please find our point-by-point responses attached.

We look forward to your decision.

Sincerely,

Diana Sonntag

Reviewer 2 Report

In this issue, Sonntag et al. describe a lifetime cost- effectiveness analysis of infant feeding modification aiming at reducing risk of later obesity, considering therefore a different aspect related to obesity, not only in term of long-term health consequences. This topic and this kind of consideration are very important in term of prevention and further analysis should be carried out.

Author Response

(The authors gave the same response as above.)

Reviewer 3 Report

This is a secondary analysis based on the data of the OSHA trial. It estimates lifetime cost savings due to use of the LP formula. It asks the question whether the long-term effects of feeding the LP formula translate into tangible cost savings. And he answer is, as expected, yes. This way the paper is a useful contribution to the current discussion of childhood obesity.

The models used are not easily followed but rely on published methods and make plausible assumptions. This reviewer has no problem with the cost modeling. But this reviewer has a problem with the "assumed willingness of society to pay" because it is not evident what these payments would be for. One can argue that the feeding of a low protein formula should incur no costs. But it is understood that for a calculation of "cost-effectiveness" there must be costs. A possible solution would be to abandon cost-effectiveness and simply present life time cost savings. Just a suggestion.

Specific comments:

Table 1S: This table has several apparent defects: There are what appears like column headers without any entries below. The table is also unduly long. In short, the table must be restructured and completed.

Figure 3: It is not clear what the two columns represent, which is largely due to the illegibility of the abscissa. What the numbers along the abscissa represent is not clear either

The associated text states that the Figure shows that "children fed LP formula spend on average 10 fewer years in overweight ... " Where is that shown in the figure?

Table 3: explain NMB and WTP

Author Response

(The authors gave the same response as above.)

Round 2

Reviewer 1 Report

Thanks for revising the paper and considered my comments.

Reviewer 3 Report

The revisions are satisfactory